# *LRP1* and *RAGE* Genes Transporting Amyloid and Tau Protein in the Hippocampal CA3 Area in an Ischemic Model of Alzheimer’s Disease with 2-Year Survival

**DOI:** 10.3390/cells12232763

**Published:** 2023-12-04

**Authors:** Ryszard Pluta, Janusz Kocki, Jacek Bogucki, Anna Bogucka-Kocka, Stanisław J. Czuczwar

**Affiliations:** 1Department of Pathophysiology, Medical University of Lublin, 20-090 Lublin, Poland; stanislaw.czuczwar@umlub.pl; 2Department of Clinical Genetics, Medical University of Lublin, 20-080 Lublin, Poland; janusz.kocki@tlen.pl; 3Department of Organic Chemistry, Faculty of Pharmacy, Medical University of Lublin, 20-093 Lublin, Poland; jacek.bogucki@umlub.pl; 4Department of Biology and Genetics, Medical University of Lublin, 20-093 Lublin, Poland; anna.kocka@tlen.pl

**Keywords:** brain ischemia, Alzheimer’s disease, CA3 area, hippocampus, *LRP1*, *RAGE*, genes, amyloid, tau protein, transport, model

## Abstract

Explaining changes at the gene level that occur during neurodegeneration in the CA3 area is crucial from the point of view of memory impairment and the development of post-ischemic dementia. An ischemic model of Alzheimer’s disease was used to evaluate changes in the expression of genes related to amyloid transport in the CA3 region of the hippocampus after 10 min of brain ischemia with survival of 2, 7 and 30 days and 12, 18 and 24 months. The quantitative reverse transcriptase PCR assay revealed that the expression of the *LRP1* and *RAGE* genes involved in amyloid transport was dysregulated from 2 days to 24 months post-ischemia in the CA3 area of the hippocampus. *LRP1* gene expression 2 and 7 days after ischemia was below control values. However, its expression from day 30 to 24 months, survival after an ischemic episode was above control values. *RAGE* gene expression 2 days after ischemia was below control values, reaching a maximum increase 7 and 30 days post-ischemia. Then, after 12, 18 and 24 months, it was again below the control values. The data indicate that in the CA3 area of the hippocampus, an episode of brain ischemia causes the increased expression of the *RAGE* gene for 7–30 days during the acute phase and that of *LRP1* from 1 to 24 months after ischemia during the chronic stage. In other words, in the early post-ischemic stage, the expression of the gene that transport amyloid to the brain increases (7–30 days). Conversely, in the late post-ischemic stage, amyloid scavenging/cleaning gene activity increases, reducing and/or preventing further neuronal damage or facilitating the healing of damaged sites. This is how the new phenomenon of pyramidal neuronal damage in the CA3 area after ischemia is defined. In summary, post-ischemic modification of the *LRP1* and *RAGE* genes is useful in the study of the ischemic pathways and molecular factors involved in the development of Alzheimer’s disease.

## 1. Introduction

Clinical and experimental studies of Alzheimer’s disease indicate, among other aspects, amyloid deposition and modification of the tau protein, as well as damage to pyramidal neurons in the CA3 area of the hippocampus, associated with learning and memory deficits [1,2,3,4,5]. The CA3 region is an input structure responsible for encoding and influencing the recall process [6]. It is currently believed that deficits in CA3–CA3 synapses occur in the early phase of Alzheimer’s disease [3] under the influence of amyloid oligomers [7]. The above neuropathological observations and indirect molecular pathways have been associated with the influence of amyloid and tau protein for many years. Despite extensive, worldwide research on the theory of the influence of amyloid and/or tau protein on the etiology of Alzheimer’s disease, the research has stalled and has not led to any final conclusions.

A large number of studies on the causes and processes leading to Alzheimer’s disease and numerous projects aimed at searching for therapies that modify the progression and development of Alzheimer’s disease have still not brought any breakthroughs. However, a growing body of literature suggests that ischemia may play a major role in driving amyloid and tau protein in the neuropathophysiology and neuropathology of Alzheimer’s disease [1,8,9,10,11,12,13,14]. In this article, we continue to describe changes in the CA3 area of the hippocampus in an ischemic model of Alzheimer’s disease. Post-ischemic CA3 neurodegeneration has been found to trigger a sequence of pathological events that can last from minutes to even a lifetime [1,15]. Firstly, the ischemic death of pyramidal neurons in CA3 is directly caused by excitotoxicity and indirectly by the neurotoxicity of folding proteins such as amyloid and tau protein [9]. Alzheimer’s disease-related amyloid and tau protein and their genes are believed to play a key role in post-ischemic progressive and irreversible neurodegeneration in the CA3 area [9]. Secondly, post-ischemic studies showed damage to the blood–brain barrier in CA3, identical to that seen in Alzheimer’s disease [16,17]. Persistent blood–brain barrier dysfunction causes the extravasation of plasma components such as amyloid and tau protein [9,17]. Thirdly, after brain ischemia, acute and chronic neuronal changes and death in CA3 are observed during long-lasting survival [1,18], with hippocampal atrophy identical to that in Alzheimer’s disease [1,2,4,18,19]. The genes of *apoptosis* (*caspase 3*), *autophagy* (*BECN1*) and *mitophagy* (*BNIP3*) are involved in neurodegenerative processes after ischemia of the CA3 area within 2–30 days of survival [20]. Studies showed that the *caspase 3* gene was significantly expressed within 7–30 days post-ischemia [20]. *BECN1* gene expression increased significantly 30 days after ischemia [20]. However, *BNIP3* gene expression was below control values for 2–30 days post-ischemia [20]. Fourthly, the neuroinflammatory response plays an important role in the progression of CA3 neurodegeneration after ischemia, such as in Alzheimer’s disease [4,15]. Proinflammatory mediators activated astrocytes and microglia in the CA3 area post-ischemia with survival for up to 2 years [4,15]. Fifth, studies of the immunoreactivity of various fragments of the amyloid protein precursor after ischemia with survival for up to 1 year showed the deposition of amyloid and the C-terminal of the amyloid protein precursor in the vicinity of blood vessels [16]. This was confirmed by gene expression in the CA3 region with survival from 2 to 30 days after ischemia, associated with the processing of the amyloid protein precursor, indicating the amyloidogenic metabolism of the precursor to amyloid [9]. The expression of *tau protein* was above control values 7–30 days after ischemia and had an impact on neuronal survival [9]. Sixth, a lack of acetylcholine in the hippocampus has been noted in post-ischemic neurodegeneration, as in Alzheimer’s disease. Acetylcholine is an essential neurotransmitter that plays a key role in synaptic transmission and memory, and its lack is a significant cause of dementia [21]. As a result of ischemia, cognitive impairment ranges from mild to severe and occurs in approximately 35–70% of people who survive one year after brain ischemia [22,23]. After ischemia, the CA3 area is involved in the occurrence of deficits in cognitive function and memory. It is estimated that after the first cerebral ischemia, approximately 13% of patients will develop Alzheimer’s disease-type dementia, and, after the second one, this rises to more than 40% [12,22]. 

The aim of this work is to continue a series of studies focusing on the quantitative determination of genes related to Alzheimer’s disease using the RT-PCR protocol, i.e., genes related to the amyloid transport *receptor for advanced glycation end products* (*RAGE*) and *low-density lipoprotein receptor-related protein 1 (LRP1*) in the CA3 area of the hippocampus, in rats that survived for 2, 7 and 30 days and 12, 18 and 24 months after an ischemic episode.

## 2. Materials and Methods

Female Wistar rats (n = 106, 53 post-ischemia and 53 controls, 2 months old, body weight 120–150 g) were subjected to 10-min global brain ischemia with survival post-ischemia of 2, 7 and 30 days and 12, 18 and 24 months [9,20]. In this work, 2.0% isoflurane carried by O_2_ was used as anesthesia [9,20]. Cerebral ischemia was caused by cardiac arrest. Details of cardiac arrest have been reported in previous studies [9,20]. After cardiac arrest lasting 10 min, the rats were resuscitated using artificial ventilation and external cardiac massage [9,20]. Sham-operated rats (control) with survival times ranging from 2 days to 24 months were subjected to the same procedures but without the induction of cerebral ischemia.

Control and post-ischemic rats were housed in pairs in cages under a 12-h light/dark cycle in rooms with humidity of 55 ± 5% and a temperature of 23 ± 1 °C. The rats had free access to tap water and laboratory chow. Experiments were always performed during the day, and rats were treated in accordance with NIH recommendations and European Community Council Directive No. 142 concerning the treatment of laboratory animals. Moreover, the Local Ethics Committee approved all procedures performed (No. 64/2010, 8 October 2010). Efforts were made to minimize rat suffering and reduce their numbers.

Before sampling the CA3 subfield, animals were perfused by the left ventricle with cold 0.9% NaCl to remove blood from the vessels. Samples with a volume of approximately 1 mm^3^ were taken from both sides of CA3 and placed in the RNAlater solution (Life Technologies, Carlsbad, CA, USA) [9,20].

The isolation of RNA was performed according to the method of Chomczyński and Sacchi [24]. RNA quality and quantity were assessed using a NanoDrop 2000 spectrophotometer (Thermo Scientific, Carlsbad, CA, USA) [9,20]. RNA was stored at −20 °C in 80% ethanol [9,20]. For reverse transcription into cDNA, 1 µg of RNA and a dedicated kit were used according to the manufacturer’s instructions (Applied Biosystems, Foster City, CA, USA). cDNA synthesis was done using the Veriti Dx apparatus (Applied Biosystems, Foster City, CA, USA) in accordance with the manufacturer’s instructions. The obtained cDNA was then amplified using real-time gene expression analysis (qPCR) on the 7900HT Real-Time Fast System (Applied Biosystems, Foster City, CA, USA) with Power Mix Master Mix SYBRgreen PCR reagent [9,20]. The number of PCR cycles in which the fluorescence level exceeded a specific relative expression threshold cycle (CT) was used in research software (Applied Biosystems, Foster City, CA, USA) to estimate the number of DNA molecules found in the mixture at the beginning of the reaction. Gene normalization was performed against the endogenous *Rpl13a* gene, the relative gene expression (RQ) was assessed based on the ΔCT method, and the results were reported as RQ = 2^−ΔΔCT^ [9,20,25]. Finally, RQ was presented after logarithmic transformation RQ (LogRQ) [9,20]. LogRQ = 0 indicated that there were no changes in the expression of the tested gene. LogRQ < 0 indicated a decrease in gene expression, and LogRQ > 0 indicated an increase in the expression of the tested gene after ischemia.

Statistica v. 12 was used to statistically evaluate the data using the non-parametric Kruskal–Wallis test with the “z” test for multiple analyses of differences between groups. Results are presented as means ± SD. *p* ≤ 0.05 was considered statistically significant.

## 3. Results

### 3.1. LRP1 Gene Expression after Ischemia

In the CA3 area, 2 and 7 days post-ischemia, *LRP1* gene expression was below control values. Two and seven days post-ischemia, the median was −0.242- and −0.379-fold change, respectively, whereas the lowest value was −0.801- and −0.556-fold change, and the maximum −0.107- and −0.201-fold change, respectively. From 1 to 24 months of survival after injury, the expression values of the tested gene systematically continued to increase above the control values. Therefore, 1 month post-ischemia, the median value was 0.116-fold change, the lowest value was 0.028-fold change, and the highest value was 0.444-fold change. After surviving for 12 months following ischemia, the median value was 0.472-fold change, the minimum value was 0.063-fold change, and the maximum value was 0.908-fold change. The maximum expression of the *LRP1* gene was recorded 18 months after ischemia, where the median value was 0.943-fold change, the lowest value was 0.127-fold change, and the highest expression value was 1.817-fold change. Two years post-ischemia, gene expression was still above control values. The median value was 0.570-fold change, the lowest value was 0.203-fold change, and the highest was 0.801-fold change. Figure 1 shows the mean values and statistically significant changes in *LRP1* gene expression. Statistically significant differences in the levels of gene expression were seen between 2 days and 12 months (z = 4.297, *p* = 0.0003); 2 days and 18 months (z = 5.194, *p* = 0.000003); 2 days and 24 months (z = 4.340, *p* = 0.0002); 7 days and 12 months (z = 4.006, *p* = 0.0009); 7 days and 18 months (z = 4.710, *p* = 0.00004); and 7 days and 24 months (z = 4.112, *p* = 0.0006) after ischemia (Figure 1). No statistically significant changes were found between 2 and 7 days; 2 and 30 days; and 7 and 30 days, or between 30 days and 12, 18 and 24 months (Figure 1).

### 3.2. RAGE Gene Expression after Ischemia

The expression of the *RAGE* gene in the CA3 subfield after ischemia on days 7 and 30 was above control values; in the remaining periods of survival, it remained below control values (Figure 2). Two days after ischemia, the median expression of the gene was −0.297-fold change, the lowest value −1.155-fold change and the highest −0.107-fold change. Seven and 30 days after the ischemic injury of the CA3 area, the median was 0.263-fold change and 0.140-fold change, respectively. However, the lowest expression values were 0.105-fold change and 0.107-fold change, respectively. The highest values were 0.655-fold change and 0.610-fold change, respectively. One year after ischemia, the median was −0.698-fold change, with the lowest value of −1.796-fold change and the highest value of −0.272-fold change. One and a half years after ischemia, the median was −0.519-fold change, with a minimum value of −0.642-fold change and a maximum value of −0.169-fold change. Two years after the CA3 ischemic episode, the median *RAGE* gene expression was −0.392-fold change, the lowest value was −0.620-fold change, and the highest value was −0.168-fold change. Figure 2 shows the changes in the mean gene expression levels and their statistical significance after survival from 2 days to 24 months. It indicates statistically significant differences in the levels of gene expression between 2 and 7 days (z = 3.992, *p* = 0.0009); 2 and 30 days (z = 3.668, *p* = 0.004); 7 days and 12 months (z = 5.097, *p* = 0.000005); 7 days and 18 months (z = 3.623, *p* = 0.004); 7 days and 24 months (z = 3.666, *p* = 0.004); 30 days and 12 months (z = 4.800, *p* = 0.00002); 30 days and 18 months (z = 3.429, *p* = 0.009); and 30 days and 24 months (z = 3.432, *p* = 0.009) post-ischemia (Figure 2). No statistically significant changes were found between 2 days and 12, 18 and 24 months, or between 12 months and 18 and 24 months and 18 and 24 months (Figure 2).

## 4. Discussion

In this investigation of the ischemic model of Alzheimer’s disease, we extended our previous studies on the deleterious effects of genes related to the metabolism of amyloid protein precursors and changes in tau protein on changes in the expression of genes related to amyloid transport (*LRP1* and *RAGE*), in order to deepen our understanding of the etiology of Alzheimer’s disease. Our current research aims to shed light on the amyloid transport of the *LRP1* and *RAGE* genes’ expression and, according to recent reports, also tau protein in post-ischemic neurodegeneration with long-term survival of up to 2 years [26,27,28].

The expression of the *RAGE* gene significantly increases during early survival after ischemia (7–30 days), but, at the late stages (12–24 months), its expression progressively and significantly decreases. With regard to *LRP1* gene expression, the situation was the opposite. In the early phase after ischemia (2–7 days), there was a significant decrease in its expression. Next, its expression increased progressively and was statistically significant in the late periods (12–24 months). In other words, in the early post-ischemic phase, the expression of the genes transporting amyloid and tau protein to the brain increases (7–30 days) with the increased neurotoxic activity, e.g., of amyloid. Conversely, in the late post-ischemic phase, amyloid clearing gene activity increases in the CA3 area, reducing and/or preventing further neuronal damage or promoting healing. 

Within 2 days post-ischemia, we observed a reduction in *RAGE* gene expression, which correlated well with the reduction in RAGE in the brain and blood after focal ischemia reported by Greco et al. [29]. According to the study of Ma et al. [30], progressive injury to hippocampal pyramidal neurons began 3 days after hypoxic–ischemic insult with marked *RAGE* mRNA expression. RAGE-positive neurons were angular with condensed nuclei. They were also positive for NeuN and were identified as dying neurons also by other stainings. Moreover, some neurons were positive for caspase 3, a marker of apoptosis [30]. RAGE immunoreactivity as well as protein levels were significantly increased in the ischemic hippocampus starting on day 3–5 after transient forebrain ischemia in gerbils [31]. Both studies indicate that the changes presented above clearly coincide with the onset of the increase in *RAGE* gene expression on day 7 in our study. These data indicate that *RAGE* is expressed in dying neuronal cells and suggest that *RAGE* may play a role in neuronal death mediated by hypoxic–ischemic injury [30,31,32].

RAGE has been shown to mediate injury in focal brain ischemia, contributing to neuroinflammation [33] and synaptic dysfunction and modulating the severity of injury in the amyloid milieu [34,35]. Furthermore, *RAGE* expression is modulated by hypoxia-inducible factor-1-α, a transcription factor activated during ischemia [36]. In particular, overexpression of the neuronal *RAGE* gene and an increase in its protein have been shown to make the brain more susceptible to ischemic injury [34]. Another study showed that the inhibition of *RAGE* signaling results in neuroprotection [36]. Our current work at least partially confirms this observation.

Previous studies have shown that RAGE is involved in disrupting synaptic function through cooperation with amyloid [35]. In light of this, it is suggested that a neuroinflammatory pathway driven by the RAGE–amyloid interaction may be the underlying mechanism linking the ischemic pathology to Alzheimer’s disease development [35]. It is already known that RAGE is closely associated with neurotoxicity and amyloid pathology [37,38]. Considering that, in Alzheimer’s disease, the expression of the *RAGE* gene is increased in various types of brain cells and it can perform many functions, it can be assumed that it influences the development of Alzheimer’s disease [39,40] as well as post-ischemic brain damage. For example, the binding of RAGE to amyloid induces neurotoxicity in neuronal cells, triggers a neuroinflammatory response in neuroglial cells and mediates the transport of amyloid across the blood–brain barrier to the brain, which enhances the pathological activity of amyloid in the brain [41,42,43]. High *RAGE* gene expression increases amyloid deposition and apoptosis in the brain, causing cognitive impairment in a mouse model of Alzheimer’s disease [44], suggesting that the same is likely true for post-ischemic brain neurodegeneration.

It is now clear that post-ischemic brain neurodegeneration is a tauopathy [9,45,46,47,48,49,50,51]. Some data suggest that in neurons and microglia, RAGE binds to the modified tau protein and facilitates the development of neuronal tau protein-related pathologies and behavioral deficits [28]. RAGE also promotes tau protein hyperphosphorylation through the activation of GSK3β [52,53]. RAGE has also been shown to be associated with the tau protein pathology by affecting the propagation of the transsynaptic tau protein in neuronal cells and inducing an inflammatory response in microglia [28]. The demonstrated induction of *RAGE* expression in neuronal cells by tau protein is believed to underlie its role in tau protein pathological propagation [28]. In contrast, the presence of amyloid provided by RAGE may serve as a trigger for the development of tau protein aggregates [54]. Amyloid accumulation in the brain may also increase *RAGE* expression and accelerate tau protein modification [55,56], serving as a key receptor in the pathogenesis of Alzheimer’s disease and highly likely in post-ischemic brain neurodegeneration. Another study indicated that the RAGE mediated amyloid accumulation in a mouse model of Alzheimer’s disease [57]. Therefore, RAGE is a potential therapeutic target in reducing the abnormal accumulation of amyloid, and this has an impact on inhibiting the progression of post-ischemic neurodegeneration of the Alzheimer’s disease type [57]. It has already been shown that blocking the AGE/RAGE signaling pathway has a beneficial effect on post-ischemic changes [58].

Additionally, previous reports have demonstrated a key role for neuronal and microglial RAGE in ischemia-induced neuronal death and neuroinflammation in permanent focal brain ischemia [59,60,61]. On the other hand, after the bilateral occlusion of the common carotid arteries, which causes global brain ischemia and consequently leads to delayed neuronal death, RAGE induction in hippocampal vascular endothelial cells precedes RAGE induction in neurons and neuroglial cells, suggesting that endothelial RAGE may cause delayed neuronal death by damage to blood vessels and consequently lead to microcirculation disorders [61,62]. Additionally, the above suggestion is confirmed by the observation that, during brain ischemia, the pyroptosis of endothelial cells occurs via the HIF-α-RAGE-NLRP3 signaling pathway, which results in irreversible damage to the microcirculation with progressive neurodegeneration [63]. It is worth adding that *RAGE* KO mice had significantly reduced neuronal cell death in the CA1 area after brain ischemia, and also had significantly reduced neuroinflammation and vascular damage [62]. Another study demonstrated a direct role for neuronal *RAGE* in promoting post-ischemic neurodegeneration in mice. The dominant-negative form of *RAGE* mice presented a reduced infarct size, indicating that RAGE signaling is directly linked to the brain ischemic pathology [34]. The activation of innate immunity (macrophages and microglia) plays a role in both the progression of post-ischemic brain neurodegeneration and Alzheimer’s disease. Cerebral ischemia is also characterized by hypoxia. Interestingly, hypoxia induces RAGE activation in macrophages [64], suggesting that RAGE may also be involved in the activation of innate immunity. In agreement, RAGE and its ligand HMGB1 have been shown to play a role in ischemic brain injury caused by infiltrating macrophages [33]. The role of *RAGE* in promoting macrophage infiltration was studied in *RAGE*-deficient mice in bone marrow cells, and this action was shown to reduce the infarct size after local brain ischemia [33]. These studies indicate that *RAGE* may at least partially mediate the effect of ischemia on the development of Alzheimer’s disease-type neurodegenerative changes.

The *LRP1* gene is expressed in neurons, astrocytes, microglia and endothelial cells [27]. There is evidence that during the survival period of 2–7 days post-ischemia, the regulated intramembrane proteolysis of LRP occurs via γ-secretase [65]. During this time, *γ-secretase* gene expression increased in our model [9] and it may cause cell death in ischemic conditions [65]. The above data correlate with the decrease in *LRP1* gene expression observed in our experiments 2–7 days after ischemia. LRP1 has been shown to directly bind amyloid and participate in its removal from the brain across the blood–brain barrier [66,67]. Furthermore, LRP1 is an endocytic receptor that transports ligands from the cell surface to the endosomal compartment, where the ligands are typically sorted into the lysosomal compartment and degraded, suggesting the dual involvement of LRP1 in Alzheimer’s disease development [27] and possibly in post-ischemic brain neurodegeneration. LRP1 has also been shown to control tau protein endocytosis and spreading [27]. The obtained experimental data identify LRP1 as a key regulator of tau protein spread in the brain and therefore a potential target for the treatment of diseases related to the spread and aggregation of this protein [26]. LRP1 has been shown to rapidly internalize tau protein, which is then efficiently degraded in the lysosomal compartment [27]. This study identified LRP1 as an endocytic receptor that binds, transports and contributes to the processing of monomeric forms of tau protein, leading to its degradation and ultimately preventing its seeding [27]. The balance of these processes may presumably be fundamental to the spread of neuropathology throughout the brain in Alzheimer’s disease [27] and possibly in post-ischemic brain neurodegeneration. Low-density lipoprotein receptor-related protein 1 has been suggested to interact with heparan sulfate proteoglycans to control tau protein entry into neuronal cells [26], but its involvement in tau protein modification has not been definitively determined.

LRP1 activation has been shown to attenuate oxidative stress, neuroinflammation and apoptosis and can improve short- and long-term neurological deficits and positively influence mortality after cerebral ischemia by inhibiting the TXNIP/NLRP3 signaling pathway [68]. The activity of the LRP1/TXNIP/NLRP3 signaling pathway was significantly elevated starting 3–5 days after ischemic brain injury [68], which correlated with the onset of the increase in *RAGE* gene expression in our study. This study also revealed that LRP1 exerted neuroprotective effects by interacting with human and mouse apolipoprotein E [68]. 

The data indicate that these two genes influence damage to the CA3 area of the hippocampus, which is an additional, previously undescribed pathological phenomenon in the form of the damage and death of neurons in this area, which contributes to damage to the function of the entire hippocampus. These data confirm the already published acute and chronic neuropathological changes in the entire hippocampus and the hippocampal CA3 area [1]. Moreover, they correlate with the previously demonstrated memory impairment and development of dementia in these rats following ischemia [69].

In conclusion, the presented data clearly indicate that in an ischemic model of Alzheimer’s disease, as demonstrated for the first time in our study, reduced *RAGE* gene expression 30 days post-ischemia likely contributes to the reduction, delay and/or transient inhibition of amyloid accumulation in CA3, and that this effect is maintained and enhanced by a parallel natural increase in *LRP1* gene expression. This points to an opposing effect of both genes after long-term post-ischemic survival, with a predominant positive effect of the *LRP1* gene. It should be emphasized that this spontaneously/naturally occurring phenomenon is currently not fully explained and further research is required on its mechanisms of action and durability. Our finding suggests that the *RAGE* gene and its protein are potential targets to limit abnormal amyloid accumulation and tau protein modification, which may have an impact in terms of inhibiting the progression of Alzheimer’s disease-type neurodegeneration after ischemia. However, further research is needed to fully understand the relationship between amyloid and its transport genes and ischemia, both early and late after ischemia. This requires the elucidation of any mechanisms at the genome and proteome level and their duration associated with the different stages post-ischemia. By focusing on the above molecular factors, the exact ischemic mechanism that triggers neuronal death would be largely unraveled in Alzheimer’s disease. We hope that the search for post-ischemic modulators will contribute to the discovery of new therapeutic drugs or preventive complementary compounds. By changing the paradigm from amyloid as the main culprit in Alzheimer’s disease to an ischemic factor, it may be possible to re-approach the etiology of Alzheimer’s disease. 

## Figures and Tables

**Figure 1 cells-12-02763-f001:**
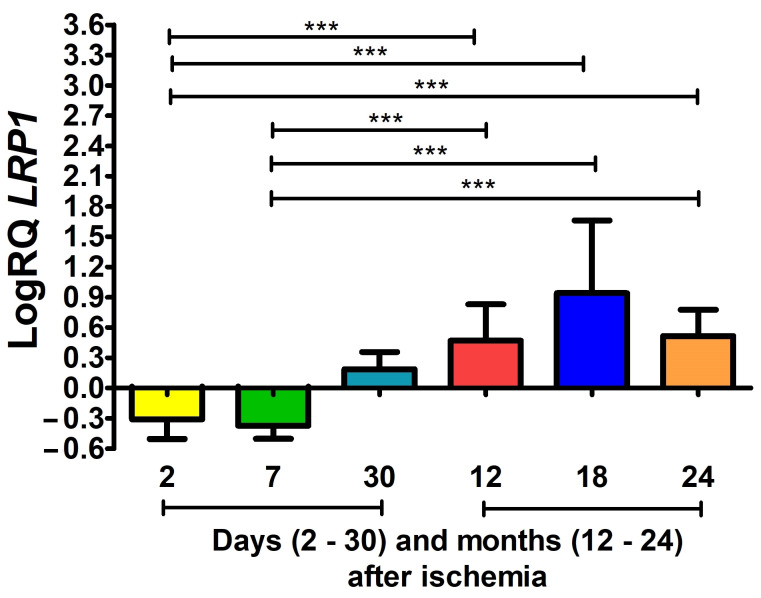
The mean *LRP1* gene expression levels in the CA3 region of the hippocampus in rats 2 (n = 10), 7 (n = 8) and 30 (n = 7) days and 12 (n = 10), 18 (n = 10) and 24 (n = 8) months after 10-min cerebral ischemia. Marked SD, standard deviation. Kruskal–Wallis test. *** *p* ≤ 0.001.

**Figure 2 cells-12-02763-f002:**
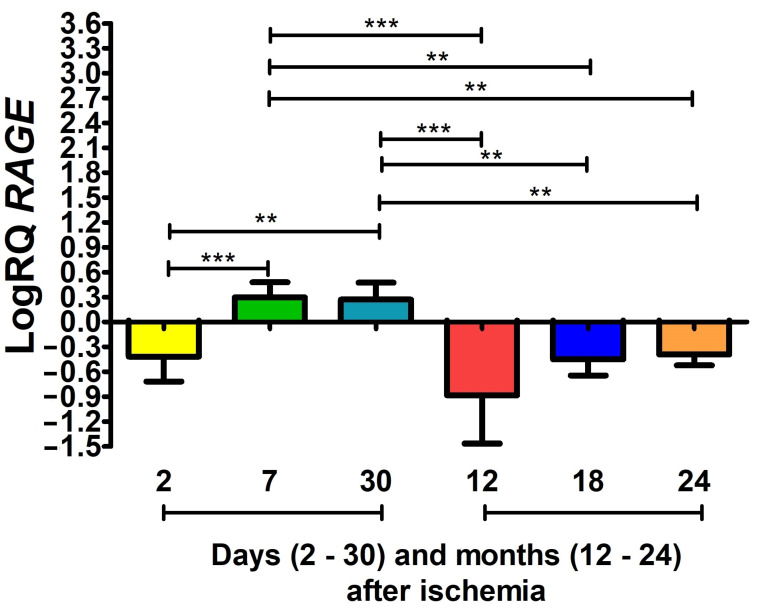
The mean *RAGE* gene expression levels in the CA3 region of the hippocampus in rats 2 (n = 10), 7 (n = 8) and 30 (n = 7) days and 12 (n = 10), 18 (n = 10) and 24 (n = 8) months after 10-min cerebral ischemia. Marked SD, standard deviation. Kruskal–Wallis test. ** *p* ≤ 0.01, *** *p* ≤ 0.001.

## Data Availability

Data are contained within the article.

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
