# Peer review of "LRP1* and *RAGE* Genes Transporting Amyloid and Tau Protein in the Hippocampal CA3 Area in an Ischemic Model of Alzheimer’s Disease with 2-Year Survival"

_cells, 2023, doi:10.3390/cells12232763_

Round 1
Reviewer 1 Report
Comments and Suggestions for Authors
The aim of this work is to continue a series of studies focusing on the quantitative 94 determination of genes related to Alzheimer's disease using the RT-PCR protocol, i.e. 95 genes related to the amyloid transport receptor for advanced glycation end products (RAGE) 96 and low-density lipoprotein receptor-related protein 1 (LRP1) in the CA3 area of the hippocampus in rats that survived 2, 7 and 30 days and 12, 18 and 24 months after an 98 ischemic episode. Cerebral ischemia was caused by cardiac arrest in female Wistar rats, 2 months old,. The presented data clearly indicate that in an ischemic model of Alz-329 heimer's disease, as demonstrated for the first time in our studies, reduced RAGE gene 330 expression after 30 days post-ischemia likely contributes to the reduction, delay and/or 331 transient inhibition of amyloid accumulation in CA3, and that this effect is maintained 332 and enhanced by a parallel natural increase in LRP1 gene expression. This points to an 333 opposing effect of both genes after long-term post-ischemic survival, with a predominant 334 positive effect of the LRP1 gene.
This is a good study that deserves publication. However, given the risk of advanced age for the AD, recommend authors in the future to use aged animals (see, DOI 10.1080/17460441.2019.1573984)
Comments on the Quality of English LanguageA study of interest to the scholars of AD research
Author Response
Reviewer 1. All changes are in red. The aim of this work is to continue a series of studies focusing on the quantitative determination of genes related to Alzheimer's disease using the RT-PCR protocol, i.e. genes related to the amyloid transport receptor for advanced glycation end products (RAGE) and low-density lipoprotein receptor-related protein 1 (LRP1) in the CA3 area of the hippocampus in rats that survived 2, 7 and 30 days and 12, 18 and 24 months after an ischemic episode. Cerebral ischemia was caused by cardiac arrest in female Wistar rats, 2 months old,. The presented data clearly indicate that in an ischemic model of Alzheimer's disease, as demonstrated for the first time in our studies, reduced RAGE gene expression after 30 days post-ischemia likely contributes to the reduction, delay and/or transient inhibition of amyloid accumulation in CA3, and that this effect is maintained and enhanced by a parallel natural increase in LRP1 gene expression. This points to an opposing effect of both genes after long-term post-ischemic survival, with a predominant positive effect of the LRP1 gene. This is a good study that deserves publication. However, given the risk of advanced age for the AD, recommend authors in the future to use aged animals (see, DOI 10.1080/17460441.2019.1573984).
We would like to thank the reviewer for his assessment and interesting comment for the future. However, taking into account that Alzheimer's disease in humans develops asymptomatically for about 10-20 years, a question arises: what do we want to demonstrate and prove in old ischemic rats? It is highly probable that Alzheimer's disease is the result of repeated ischemic (so-called silent) episodes. The matter is currently open and requires deep consideration, considering that a rat can survive up to 2 years in good conditions. After this period, it will not be possible to further observe chronic changes in post-ischemic brain.
Reviewer 2 Report
Comments and Suggestions for Authors
The manuscript by Pluta et al "LRP1 and RAGE genes transporting amyloid and tau protein in 2 the hippocampal CA3 area in an ischemic model of Alzheimer's 3 disease with 2-year survival" is interesting. Overall, the manuscript is fine, however, an image of the ischemic hippocampus would be nice to show the amount of damage caused by the ischemia episode. Further, the last sentence of the discussion needs to be rewritten, it is currently not clear, "repealed"?
Author Response
Reviewer 2. All changes are in red. The manuscript by Pluta et al "LRP1 and RAGE genes transporting amyloid and tau protein in the hippocampal CA3 area in an ischemic model of Alzheimer's disease with 2-year survival" is interesting.
Thanks.
Overall, the manuscript is fine, however, an image of the ischemic hippocampus would be nice to show the amount of damage caused by the ischemia episode.
Detailed neuropathological changes in the hippocampus have already been published and are cited in the manuscript (Pluta et al., 2009). 1.Pluta, R.; Ułamek, M.; Jabłoński, M. Alzheimer’s mechanisms in ischemic brain degeneration. Anat. Rec. 2009, 292, 1863–1881.
Original text fragment and fragments added in the manuscript that refer to neuropathological changes in the entire hippocampus and in the CA3 area of the hippocampus are marked in red.
Further, the last sentence of the discussion needs to be rewritten, it is currently not clear, "repealed"?
We have removed this sentence.
Reviewer 3 Report
Comments and Suggestions for Authors
The authors examined the changes in gene expression levels related to amyloid production and clearance in the CA3 region of the rat brain following ischemic injury over time, providing some clues for the treatment of neuronal damage after ischemia. However, according to the reviewer, there are still many areas that need improvement. Here are the concerns from the reviewer:
1. In the present study, the authors selected two genes, LRP1 and RAGE, involved in amyloid transport. Why these two genes were selected and how about the expression of other related genes, like APP, PSEN1 and 2, APOE, CLU, which seems to be more relevant to the amyloid metabolism than the selected two? As the authors stated that the expression level of these two genes are crucial in memory impairment and the development of post-ischemic dementia, have the authors evaluated the memory ability and the dementia of these ischemia rat? Moreover, have the authors done any analysis to prove the crucial role of these two genes for memory and dementia development in the present study? Please clarify.
2. Real-time PCR was performed in the present study to quantify the gene expressions of LRP1 and RAGE in CA3 region. However, there are other detection methods available which are more precise than real-time PCR, like RNA sequencing, microarray analysis, single-cell RNA sequencing. In comparison to real-time PCR, these methods offer advantages in terms of sensitivity, specificity, and especially the ability to analyze a large number of genes simultaneously. Please justify why real-time PCR, instead of these more precise methods, were performed.
Comments on the Quality of English LanguageThe manuscript is well-written.
Author Response
Reviewer 3. All changes are in red. The authors examined the changes in gene expression levels related to amyloid production and clearance in the CA3 region of the rat brain following ischemic injury over time, providing some clues for the treatment of neuronal damage after ischemia. However, according to the reviewer, there are still many areas that need improvement. Here are the concerns from the reviewer:
We are very sorry that we cannot agree with the reviewer, but the changes presented in the manuscript do not concern the production of amyloid, but its transport from the blood to the brain and from the brain to the blood.
- In the present study, the authors selected two genes, LRP1 and RAGE, involved in amyloid transport. Why these two genes were selected and how about the expression of other related genes, like APP, PSEN1 and 2, APOE, CLU, which seems to be more relevant to the amyloid metabolism than the selected two? As the authors stated that the expression level of these two genes are crucial in memory impairment and the development of post-ischemic dementia, have the authors evaluated the memory ability and the dementia of these ischemia rat? Moreover, have the authors done any analysis to prove the crucial role of these two genes for memory and dementia development in the present study? Please clarify.
We have initially selected 18 Alzheimer's disease-related genes that we are working on in 8 brain structures, e.g. the CA1 and CA3 areas of the hippocampus, which we consider to be important in the development of Alzheimer's disease. Genes related to amyloid transport and genes related to amyloid precursor protein metabolism speak of different phenomena. We decided that genes presenting other phenomena should be presented separately and with an extended description. Regarding other genes inquired by the reviewer, i.e. APP, PSEN1 and 2, APOE, CLU, BNIP3, BECN1 and others are of interest to us. The expression of these genes after survival for 2-30 days has been published and is presented in the manuscript, these are references no. 10, 26. We are currently working on the remaining genes with survival times after ischemia up to two years in different brain structures.
In this article, we do not claim that the RAGE and LRP1 genes "are crucial in memory impairment and the development of post-ischemic dementia". We show that they affect the damage to the CA3 area of the hippocampus, which is an additional phenomenon that has not yet been demonstrated in the damage and death of neurons in this area and contributes to the damage to the function of the entire hippocampus. We assessed memory and dementia in these ischemic rats and we included this information in the discussion (Kiryk et al., 2011). But on condition that the assistant editor, Mr Dixon Liu, does not order the citation to be removed, because it will be self-citation. We are sorry, but nowhere in the entire manuscript do we claim that these two genes presented play a “crucial role” in the development of memory disorders and dementia, but we suggest that they play an additional role in the very complex process of post-ischemic hippocampal injury.
-Kiryk A, Pluta R, Figiel I, Mikosz M, Ulamek M, Niewiadomska G, Jablonski M, Kaczmarek L.Transient brain ischemia due to cardiac arrest causes irreversible long-lasting cognitive injury. Behav Brain Res. 2011, 219, 1-7.
- Real-time PCR was performed in the present study to quantify the gene expressions of LRP1 and RAGE in CA3 region. However, there are other detection methods available which are more precise than real-time PCR, like RNA sequencing, microarray analysis, single-cell RNA sequencing. In comparison to real-time PCR, these methods offer advantages in terms of sensitivity, specificity, and especially the ability to analyze a large number of genes simultaneously. Please justify why real-time PCR, instead of these more precise methods, were performed.
Each of the mentioned methods (like RNA sequencing, microarray analysis, single-cell RNA sequencing) has limitations in sensitivity and specificity.
Test sensitivity is the ratio of true positives to the sum of true positives and false negatives. A sensitivity of 100% for a medical test would mean that all people with the disease or generally with the specific disorder being sought would be recognized. The concept is interpreted as the ability of a test to correctly diagnose a disease where it occurs. The specificity of a test is the ratio of true negative results to the sum of true negative and false positive results. A specificity of 100% would mean that all healthy people in the performed diagnostic test would be marked as healthy. A test with high specificity has a low type I error [1, 2, 3, 4].
The real-time PCR method we use is a recognized method in this type of scientific publications. The questions raised by the reviewer were important some time ago - now no one doubts the quality of the real-time PCR method. It is considered the “gold standard” in gene expression research. In the PubMed database, it was used in 231,328 publications from peer-reviewed journals from the JCR list. Gong et al.'s 2017 analysis of mRNA-protein correlation co-occurrence in a single cell showed that many genes had a significant correlation between protein products and their transcripts. This confirms the validity of using the real-time PCR technique in our publication. The PCR technique is validated at every stage, which ensures reliable results. The use of commercial probes and other reagents ensures the repeatability of the results and significantly increases the sensitivity and specificity of our tests [5].
- Hsu JC, Chang J, Wang T, Steingrímsson E, Magnússon MK, Bergsteinsdottir K. Statistically designing microarrays and microarray experiments to enhance sensitivity and specificity. Brief Bioinform. 2007, 8(1), 22-31.
- Mi Z, Zhongqiang C, Caiyun J, Yanan L, Jianhua W, Liang L. Circular RNA detection methods: A minireview. Talanta. 2022, 238(Pt 2), 123066.
- Picelli S, Faridani OR, Björklund AK, Winberg G, Sagasser S, Sandberg R. Full-length RNA-seq from single cells using Smart-seq2. Nat Protoc. 2014, 9(1), 171-81.
- Miquel Porta: A Dictionary of Epidemiology. Oxford: International Epidemiological Association – Oxford University Press, 2008.
- Gong H, Wang X, Liu B, Boutet S, Holcomb I, Dakshinamoorthy G, Ooi A, Sanada C, Sun G, Ramakrishnan R. Single-cell protein-mRNA correlation analysis enabled by multiplexed dual-analyte co-detection. Sci Rep. 2017, 7(1):2776.
Round 2
Reviewer 1 Report
Comments and Suggestions for Authors
The authors have successfully addressed my comments
Comments on the Quality of English LanguageGood paper, likely to attract a lot of interest
Author Response
Thanks.
Reviewer 3 Report
Comments and Suggestions for Authors
No further comments.
Comments on the Quality of English LanguageNeeds to be improved.
Author Response
English corrected by a native speaker and scientist from the USA.
The number of own citations in the manuscript decreased by 7 positions.